# Systematic Comparison of Temperature Effects on Antibody Performance via Automated Image Analysis: A Key for Primary Ciliary Dyskinesia Diagnostic

**DOI:** 10.3390/cells14161236

**Published:** 2025-08-11

**Authors:** Hanna Przystalowska-Maciola, Malgorzata Dabrowska, Ewa Zietkiewicz, Zuzanna Bukowy-Bieryllo

**Affiliations:** Institute of Human Genetics Polish Academy of Sciences, Strzeszynska 32, 60-479 Poznan, Poland; hanna.przystalowska-maciola@igcz.poznan.pl (H.P.-M.); malgorzata.dabrowska@igcz.poznan.pl (M.D.); ewa.zietkiewicz@igcz.poznan.pl (E.Z.)

**Keywords:** immunofluorescence, airway epithelium, primary ciliary dyskinesia, diagnostics, automated image analysis, cilia structure

## Abstract

Immunofluorescence (IF) microscopy of ciliated epithelium is gaining increased popularity as a pre-genetic diagnostic method in primary ciliary dyskinesia (PCD). Ensuring reliable IF-based diagnostics in PCD requires robust standardization of staining methods and antibody performance. We applied whole slide scanning and automated image analysis to systematically evaluate the influence of various sample storage conditions on the specificity of IF staining. We tested eight polyclonal antibodies targeting diverse axonemal protein epitopes, routinely used for PCD diagnostics, under seven different temperature and time combinations. The storage conditions simulated handling of epithelial brushing on glass slides: after material collection at the clinic, during transport, or after reception at the diagnostic laboratory. Our study revealed that proper slide storage conditions are essential for the reliable PCD diagnosis via IF staining. We suggest continuous storage at −80 °C or −20 °C for slides prepared at the diagnostic laboratory, and storage at −20 °C or 4 °C for slides prepared remotely and shipped. Moreover, the IF sensitivity to slide storage conditions differs among antibodies targeting various ciliary elements, with molecular ruler proteins being particularly sensitive to prolonged storage at room temperature. We emphasize the inclusion of additional control slides to mitigate the inter-individual differences and the crucial correlation of IF results with comprehensive patient clinical history for enhanced diagnostic reliability.

## 1. Introduction

The majority of the human respiratory tract is lined with the specialized airway epithelium, composed of several cell types, including multiciliated cells, which carry hundreds of motile cilia on their apical surface [1,2]. Motile cilia, whose main role is to transport fluid and mucus in the organism, are microtubule-based organelles, composed of the axoneme anchored in the cell membrane by the basal body [2,3]. The ciliary axoneme is composed of nine microtubular (MT) doublets surrounding a central pair (CP) complex. Peripheral doublets are connected to the CP complex by radial spokes (RSs), and between each other by nexin–dynein regulatory complexes (N-DRC). Along the axoneme length, peripheral MT doublets of the motile cilia are decorated with multiprotein complexes, including inner (IDA) and outer (ODA) dynein arms responsible for generating and shaping the ciliary beating. The proper spatial arrangement of IDAs, RSs, and N-DRC along the axoneme depends on the “molecular ruler” (MR) complex [3].

The function of motile cilia is driven by the coordinated action of their multiprotein complexes. The CP complex, containing SPEF2 protein, acts as the central command center, controlling the asymmetry and direction of the ciliary beat. The signal is then transmitted to the peripheral MT doublets via the RSs, containing proteins like RSPH4A and RSPH9; RSs ensure the proper ciliary waveform. The N-DRC, with proteins like GAS8, further modulates the activity of the dynein arms and provides structural stability of the cilium.

The ciliary beat itself is generated by the dynein arms. The ODAs, which include the core ATPase protein DNAH5, generate the power stroke that drives the main ciliary movement. The IDAs, containing, among others, DNALI1, are involved in the fine-tuning of the ciliary beat, giving it its characteristic shape.

Finally, while not directly involved in the beating process, the MR complex is essential for proper ciliary functioning. The complex, containing CCDC39 and CCDC40, defines the precise spatial arrangement of the ODA, IDA, and RS complexes along the microtubules during ciliary assembly.

Inherited defects of the motile cilia function lead to a rare, multisystemic disease, primary ciliary dyskinesia (PCD), with the occurrence estimated between 1:10,000 and 1: 20,000 [4,5]. PCD results from changes in the structure or number of cilia and presents with several symptoms, including chronic upper and lower respiratory tract infections, abnormal mucociliary clearance, recurrent otitis media, infertility, and randomization of organ laterality in 50% of cases [2,3,6].

Genetically, PCD is highly heterogeneous. It is most frequently inherited in an autosomal recessive manner, and, rarely, in the X-chromosomal or autosomal dominant manner. Hundreds of pathogenic variants have been identified so far in over 50 different genes [3,7,8]. In spite of significant developments in the field of PCD genetics, successful detection of underlying defects is presently achieved in only 60–70% of patients [8].

Due to unspecific symptoms, PCD diagnosis is often delayed and complicated [9]. The first-line diagnostic tests in PCD, which precede identification of pathogenic genetic variants, include the following: (1) nasal nitric oxide (nNO) measurement, (2) ciliary motility assessment using high-speed videomicroscopy analysis (HSVMA), (3) analysis of motile cilia structure using transmission electron microscopy (TEM), and (4) immunofluorescent (IF) microscopy of the respiratory epithelium cells allowing visualization of ciliary proteins [2,9,10].

IF microscopy is gaining increased popularity as a pre-genetic diagnostic method in PCD [11,12,13,14]. By revealing the absence or mislocalization of specific proteins—markers of the ciliary ultrastructure elements (such as, e.g., DNAH5, DNALI1, RSPH4A, RSPH9, CCDC39, CCDC40, GAS8, and SPEF2)—it can confirm an axonemal defect in the material collected from a patient [15]. Although not always indicating the defective protein itself, it can be used to guide genetic testing, rendering it more specific and cost-effective [12,16]. An important practical advantage of the IF-based analysis is that collecting samples (typically, nasal epithelium brushing spread and dried on glass microscope slides, further referred to as slides) can be performed by medical personnel with no previous experience in PCD, and shipped to specialized diagnostic laboratories [13].

Efficient and reliable IF-based diagnostics in PCD require standardization of staining methods and antibodies. One of the challenges in the IF-based diagnostics concerns the detection of genetically determined defects in the ciliary protein synthesis or assembly, which result in weakening of the IF signal in cilia, accompanied by the presence of a non-specific diffused signal in the cytoplasm. We hypothesized that similar patterns may be observed when the quality of the analyzed material is compromised. Obviously, the nasal epithelium brushing quality depends on the donor’s health status (inflammation and infection can obscure the diagnosis [16]), but the conditions of the slide storage are also very important. So far, no study has been performed to quantify and formally test the influence of slide storage conditions on the specificity of IF staining.

The aim of the study was to compare the reliability of IF staining with selected polyclonal antibodies routinely used for PCD diagnostics at a range of slide storage conditions, chosen to reflect realistic scenarios of slide handling between sample collection and the IF analysis. The results of the study provide clues concerning proper slide handling, adherence to which should improve the reliability of the IF-based PCD diagnostics.

## 2. Materials and Methods

### 2.1. Nasal Samples

Nasal specimens from 5 non-PCD donors (average age: 36.4 ± 6.05 yr, without airway infection 4 weeks prior to collection) were collected using a cytological brush and suspended in RPMI1640 medium (ThermoFisher Scientific, Waltham, MA, USA), as before [17]. A total of 40 µL of cell suspension was pipetted onto a circle of an uncoated cytoslide (309-100-0; Tharmac GmbH, Limburg an der Lahn, Germany). Slides dried at RT overnight were stored under different conditions prior to IF analysis (Table 1). The study was conducted according to the guidelines of the Declaration of Helsinki and approved by the Ethics Committee of Poznan University of Medical Sciences (381/22). Written informed consent was obtained from all participants.

### 2.2. Immunofluorescence Microscopy

Samples were fixed and stained as before [17]; for the details, see the Appendix A. Each slide was co-stained with one of the primary rabbit antibodies and the mouse AcTub antibody (Appendix A). Images of a minimum of 30 cells were acquired using the tile scan function in LAS AF 2.7.3 software, under HC PL APO CS2 100 ×/1.4 OIL objective in the Leica DMi8 confocal microscope (Leica Microsystems GmbH, Wetzlar, Germany). Raw images were exported to TIFF images, with each channel saved as a separate file.

### 2.3. Image Analysis

Exported TIFF images were analyzed using a custom pipeline created in CellProfiler software v. 4.2.7 [18] (main pipeline steps in Figure 1). All measurements in the CP pipeline are expressed in pixels; to express values in µm, the pixel values were multiplied by the resolution (0.1136 µm/pixel).

Custom R scripts were used to analyze the parameters significantly different (*p* < 0.001) between all samples and to combine all mCilia files for all antibodies and all persons. Initial analysis of the final data was performed using pivot tables in Excel, and statistics and heatmaps were performed using GraphPad Prism v.9.5.1. A lack of unified acquisition settings (intensity) of the laser did not allow us to analyze the correlation between the intensity of the red/green signal, depending on the temperature conditions. See the Appendix A and Methods for further description of the Cell Profiler analysis and presented parameters.

## 3. Results

Slides with nasal brushings from five healthy donors were stored at seven time and temperature combinations (Table 1). Storage at −80 °C for 28 days was assumed to represent the best storage conditions. The effect of storage at an alternative temperature of −20 °C was tested for 28 days or 8 weeks. To mimic shipping of slides between clinics and diagnostic laboratories, three more conditions were examined: −20 °C, 4 °C, or RT for 11 days (clinics), followed by 3 days at RT (transport), and 14 days at −80 °C (lab). Slides stored for 28 days at RT were considered detrimental (the most unfavorable of the tested storage conditions).

After the end of the storage period, slides were stained with eight polyclonal rabbit antibodies commonly used in PCD diagnostics (see Appendix A); the secondary anti-rabbit IgG antibody staining was visualized in the red channel. Antibodies were selected to target proteins representing various elements of the axoneme, and throughout the text are referred to using targeted protein names (DNAH5–ODA, DNALI1–IDA, RSPH4A and RSPH9–RS, GAS8–N-DRC, CCDC39 and CCDC40–MR, SPEF2–CP complex). Each slide was also treated with a monoclonal mouse antibody against acetylated α-tubulin (Ac-αTub), a component of the ciliary MT; the secondary anti-mouse IgG antibody staining was observed in the green channel. The Ac-αTub staining was clearly visible and specific across the analyzed conditions, and served as a stable marker of the ciliary axoneme. An overlap of the red signal with the green marker indicated the presence of the target protein in cilia.

The specificity of staining using tested polyclonal antibodies was highly variable depending on the storage time and temperature; the size of the “storage effect” depended on the targeted ciliary proteins. Representative images of the IF staining of slides stored at the best versus the most detrimental conditions are shown in Figure 1.

### Image Analyses Using CellProfiler

In order to provide a quantitative measure of the slide storage conditions’ impact on the IF staining, automated image analysis was performed using a custom CellProfiler pipeline (Figure 2 and Appendix A).

In total, 280 slides from 5 donors across 56 combinations of storage conditions (7) and antibodies (8) were analyzed. For each combination, the values of the parameters analyzed in CellProfiler were presented as the means of individual results obtained for five tested donors. In total, 6167 images, typically representing 1–5 cells, were analyzed (on average, 22 images per slide; more detailed information below and in the Appendix A).

Image analysis in the CellProfiler pipeline returned two main categories of parameters: those that characterize objects identified within each channel and those that describe the relations between the channels. The first group of parameters measured objects’ size, shape, and fluorescence intensity distribution, while the second group assessed general channel correlation and the distance between paired objects from the same cell. All the parameters confirmed unchanged stability of the green objects (Ac-αTub staining) across various storage conditions; the green color (marker of the axoneme) was therefore used as a reference.

Among 114 parameters analyzed in CellProfiler, 81 significantly correlated with slide storage conditions (*p* < 0.001, Appendix A). The majority of them characterized red channel objects, specifically their area and shape (29 parameters) and intensity (43 parameters). The second group of parameters characterized the relationships between the red and green channels. These included seven parameters that estimated the correlation between the channels and one that measured the distance between the centroids of paired objects (Appendix A).

For further presentation, representative parameters from each parameter group were chosen. For each of the selected parameters, a heatmap was prepared, displaying the mean value averaged across slides from five individuals. The heatmaps illustrate the impact of 56 combinations of storage conditions and tested antibodies. To ensure the objects being analyzed were from the same cell, we only included paired objects in our parameter analysis.

## 4. Channel Parameters

### 4.1. Area Characteristics

The impact of slide storage conditions on the area, eccentricity, and compactness of the objects in the red channel was compared for various tested antibodies (Figure 3). At a given storage condition, the parameters’ values indicating cilia-specific staining using a tested antibody were most similar to those characterizing a stable axoneme marker: low object’s area and low compactness (indicating shape regularity), as well as high eccentricity (range 0–1, higher values indicating stretched rather than circular shape).

For antibodies targeting RSPH4A, RSPH9, DNAH5, or GAS8, the area size and compactness values were low (similar to axonemal marker) across various slide storage conditions (Figure 3A,B). Antibodies targeting CCDC39, CCDC40, SPEF2, and DNALI1 displayed lower specificity of staining (higher object area and compactness) for slides stored at less favorable conditions, especially those including prolonged storage at RT. Eccentricity values yielded less consistent results. High eccentricity values (similar to the axoneme marker, indicating high staining specificity) were observed for RSPH4A across various conditions, even those including prolonged times at RT that were detrimental to staining with other antibodies (Figure 3C). Surprisingly, prolonged slide storage at −20 °C resulted in low eccentricity values for staining with RSPH9, GAS8, and DNAH5.

### 4.2. Intensity Parameters

The intensity mass displacement parameter characterizes the distribution of the signal within the object. It measures the distance between the intensity-weighted center of the object and its geometric center: the higher the distance, the more asymmetric is the signal intensity distribution within the object.

The calculated intensity mass displacement values indicated various levels of red signal distribution within the cells (Figure 3D) depending on the storage conditions. Prolonged storage at RT (RT_28 d, RT/RT/−80) resulted in a marked increase in this parameter, suggesting that the red signal was broadly distributed within the cell, without accumulation in cilia (Figure 3D). This is in agreement with the IF images, where storage at RT_28 d (most detrimental conditions), compared to −80_28 d (best conditions), caused a visible reduction in the red staining within the cilium, and an increase in the red staining in the cytoplasm (Figure 1). At storage conditions, which did not involve prolonged incubation at RT, the lowest red signal asymmetry was observed for RSPH4A, RSPH9, DNAH5, and GAS8 proteins, while for SPEF2, DNALI1, CCDC39, and CCDC40, the asymmetry was strikingly higher.

### 4.3. Analysis Between Channels

#### 4.3.1. Pairing of Tested Antibodies with the Axoneme Marker

Among 6167 analyzed images, the CellProfiler pipeline detected 31,566 objects in the red channel, of which 23,347 (~74%) at least partially overlapped with objects identified in the green channel (axoneme marker). Note that the percentage of unpaired objects may include both unpaired red and green objects within the same cell and objects belonging to different cells.

#### 4.3.2. Correlation Characteristics

The RED–GREEN correlation characterizes the correlation between the pixel intensities of the red channel and the green channel. It is an image-based (as opposed to object-based) parameter, which assesses how changes in the red signal intensity are linked to changes in the green signal at the same spatial locations within the image. The values of the RED–GREEN correlation can range from −1 to 1. High positive values indicate that when the red is high (or low), the signal in the green channel tends to be high (or low) as well; values lower than 0 suggest a reverse correlation.

The RED–GREEN correlation was at the top of the list of the parameters significantly correlating with storage conditions (it had the smallest *p*-value of 2.30 × 10^−50^) (Appendix A). Of note, the GREEN–RED correlation parameter (describing the opposite correlation between the localization and intensity of the green signal with that of the red signal) was not significantly associated with storage conditions (*p*-value 0.75) (Appendix A).

Analysis of the overall RED–GREEN correlation values (averaged across all tested antibodies) confirmed that storage at RT for longer times (RT/RT/−80 and RT_28 d) resulted in a considerable reduction in this parameter (Figure 3E).

Apart from these longer RT storage conditions, the RED–GREEN correlation values were usually rather stable for antibodies targeting RS, ODA, and CP (RSPH4A, RSPH9, DNAH5, and SPEF2). Antibodies targeting MR and IDA proteins (CCDC39, CCDC40, DNALI1) displayed less consistent results; the lowest RED–GREEN correlation values across different storage conditions were observed for CCDC39 (Figure 3E).

#### 4.3.3. Distance Between the Object Pairs

Distance_Centroid_mCilia parameter measures the distance between the geometric centers (centroids) of the objects in the red channel (tested antibodies) and green channel (axoneme marker). The differences in the area and shape between the objects in the analyzed channels observed under different storage conditions, was also reflected in considerable differences in the Distance_Centroid_mCilia parameter (Appendix A).

Heatmap comparison of the distances for each tested antibody and storage conditions is shown in Figure 3F. For CCDC39, even at the best conditions, the average distances between the signal and axoneme marker were significantly higher (6.07 ± 0.73 µm vs. 3.56 ± 0.83 µm, *p*-value < 0.0001) than for all other tested antibodies (Figure 3F).

#### 4.3.4. Integrated Performance Metrics

The ranks obtained for each of the analyzed parameters were averaged into one metric, which assessed optimal storage conditions across all tested antibodies (Table 2) and the general performance of each tested antibody averaged across the storage conditions (Table 3).

The best storage conditions (Table 2, green) were −80_28 d and −20_28 d. Increased time of storage at −20 °C caused a slight drop in the performance of some of the antibodies. Storage conditions, including the short-term storage at RT (“shipping”), were less favorable, although the RT incubation only for 3–4 days did not considerably influence IF results (Table 2, yellow). The two definitively worst storage conditions were those including long-term storage at RT(RT_28 d, RT/RT/−80) (Table 2, red).

The three best-performing antibodies were those targeting RSPH4A, DNAH5, and GAS8, while antibodies against MR proteins and DNALI1 were characterized by the worst performance (Figure 3; Table 3). The poor performance of the latter antibodies appeared to be caused by their large sensitivity to long storage at RT (Figure 3). However, even at storage conditions excluding prolonged RT, antibodies targeting MR proteins and DNALI1 performed worse than other antibodies (Table 4, Appendix A). Interestingly, GAS8, which was relatively insensitive to the prolonged RT, at other conditions performed worse than RSPH4A and DNAH5 (Figure 3).

#### 4.3.5. Color Switch of the Antibodies 

In order to assess the specificity of the observations, we have compared the results of the IF stainings for DNALI1 antibody stored at -80oC for 28d, when the fluorochromes associated with secondary antibodies were switched to opposite colors (axoneme marker –red, DNALI1-green). Although the change for DNALI1_AF488 was associated with an increase /or decrease in the absolute parameter values, the relative differences between the positive and negative control were similar (Appendix A). 

## 5. Discussion

We applied automated image analysis to systematically examine the influence of slide storage conditions on the specificity of IF staining with antibodies routinely used for PCD diagnostics. In an attempt to identify the best storage conditions applicable in the clinical practice, we have tested various combinations of temperatures and storage times, which mimic the procedures of handling epithelial brushing on glass slides: at the clinic collecting the material, during transport, or after reception at a diagnostic laboratory.

### 5.1. Slide Storage Condition

The analysis of slide storage conditions indicated that both storage temperature and time were important factors influencing IF staining specificity.

As expected, the long-term (>14 days) storage at RT had the worst impact on the IF staining specificity, as evidenced by the lowest overall performance scores (Table 2). For the majority of parameters tested, four weeks of storage at RT, or two weeks of storage at RT followed by two weeks at −80 °C, yielded the worst values of tested parameters (Figure 1 and Figure 3, Table 2).

Uninterrupted 4-week storage at −80 °C, or −20 °C, had the least impact on the staining by tested antibodies; prolongation of the storage at −20 °C to 8 weeks did not lead to worse staining results.

Comparison of the results for conditions containing an intermittent 3 d RT incubation period, mimicking transport of slides, indicates that the short-term increase in the temperature had no remarkable effect on the parameters describing the specificity of IF staining (Figure 3, Table 2). 

Our results indicate that samples to be shipped from the clinics to diagnostic laboratories should be stored at −20 °C or 4 °C; as samples are not affected by a short incubation at RT, the shipment can be completed at RT. After reception at a diagnostic laboratory, samples should be stored at −80 °C until IF analysis is performed (Figure 3, Table 2).

### 5.2. Epitope Sensitivity to Storage Conditions

Specific IF detection of various ciliary complexes using antibodies targeting RSs (RSPH4A, RSPH9), ODA (DNAH5), N-DRC (GAS8), and CP protein (SPEF2) was relatively insensitive to various slide storage conditions, in the case of RSPH9 and GAS8, including also a long-term exposure to the RT. The staining specificity of IDA protein (DNALI1) antibody was very vulnerable to the prolonged slide storage at RT, but even more so to the repeated freeze–thawing process (compare various storage conditions in Figure 3). Of all antibodies, CCDC39 staining was the most sensitive to storage conditions, and satisfactory results with this antibody were observed only when slides were stored at −80 °C (Figure 1 and Figure 3).

IF detection of both MR proteins (CCDC39 and CCDC40) was negatively affected by almost all slide storage conditions (Table 3 and Table 4, Appendix A), rendering CCDC39 or CCDC40 antibodies unsuitable in an IF-based search for MR defects. Fortunately, these defects have been previously shown to be detectable by IF staining using GAS8 antibody [14,19,20,21], which in our study was storage condition-resistant (Table 3 and Table 4, Appendix A). However, if the use of the CCDC39 antibody is, for some reason, necessary, dried slides should be frozen at −80 °C as soon as possible and stored at this temperature without any interruption.

### 5.3. Interindividual Differences

Relatively large inter-individual differences, not associated with donors’ age or sex, were observed (example in Appendix A). For example, even at optimal storage conditions, diffused staining (red present both in cilia and in the cytoplasm) of otherwise stable ciliary proteins was consistently observed in slides from Donor 4 (Appendix A). Importantly, ALI culture differentiation of the airway epithelium from Donor 4 increased proper ciliary localization of the red signal (Appendix A), suggesting that the stability of antibodies’ epitopes had been compromised by environmental factors (like in secondary ciliary dyskinesia) [22,23]. Nevertheless, the possibility of a nonspecific cytoplasmic staining even in properly stored slides from healthy donors raises concerns, as it may obscure IF-based PCD diagnostics in cases where pathogenic variants cause diffused cytoplasmic location of the ciliary protein rather than its complete lack from cilia [14,24]. ALI culture, erasing possible environmental effects, seems to alleviate this problem. However, when IF-based diagnostics are to be made using uncultured airway epithelium samples, it is essential to properly store the slides and to take note of the patient’s health status at the time of material collection.

### 5.4. Limitations of the Study

The low number of individuals tested did not allow us to more precisely assess inter-individual differences. Only selected storage conditions and rabbit primary antibodies were tested. We assume that the observed deterioration of the IF staining at some storage conditions reflected targeted epitope instability. Moreover, differences in the performance of particular antibodies (even at “optimal” conditions) could reflect antibody polyclonality. Therefore, it is possible that different epitopes of the same ciliary protein have different sensitivity for storage conditions, and using antibodies targeting other parts of the protein would improve the IF performance.

## 6. Conclusions

Our study has revealed that proper conditions of the slide storage before IF staining are essential for the reliable PCD diagnosis via specific IF staining. If the microscopic slides are prepared at the diagnostic laboratory, we suggest continuous storage at −80 °C or −20 °C. If the slides are prepared at a collaborating clinic and shipped, we suggest storage at −20 °C until shipment. In any case, the possibility of inter-individual differences should be kept in mind, and IF results indicative of PCD should always be confronted with the patient’s clinical history.

## Figures and Tables

**Figure 1 cells-14-01236-f001:**
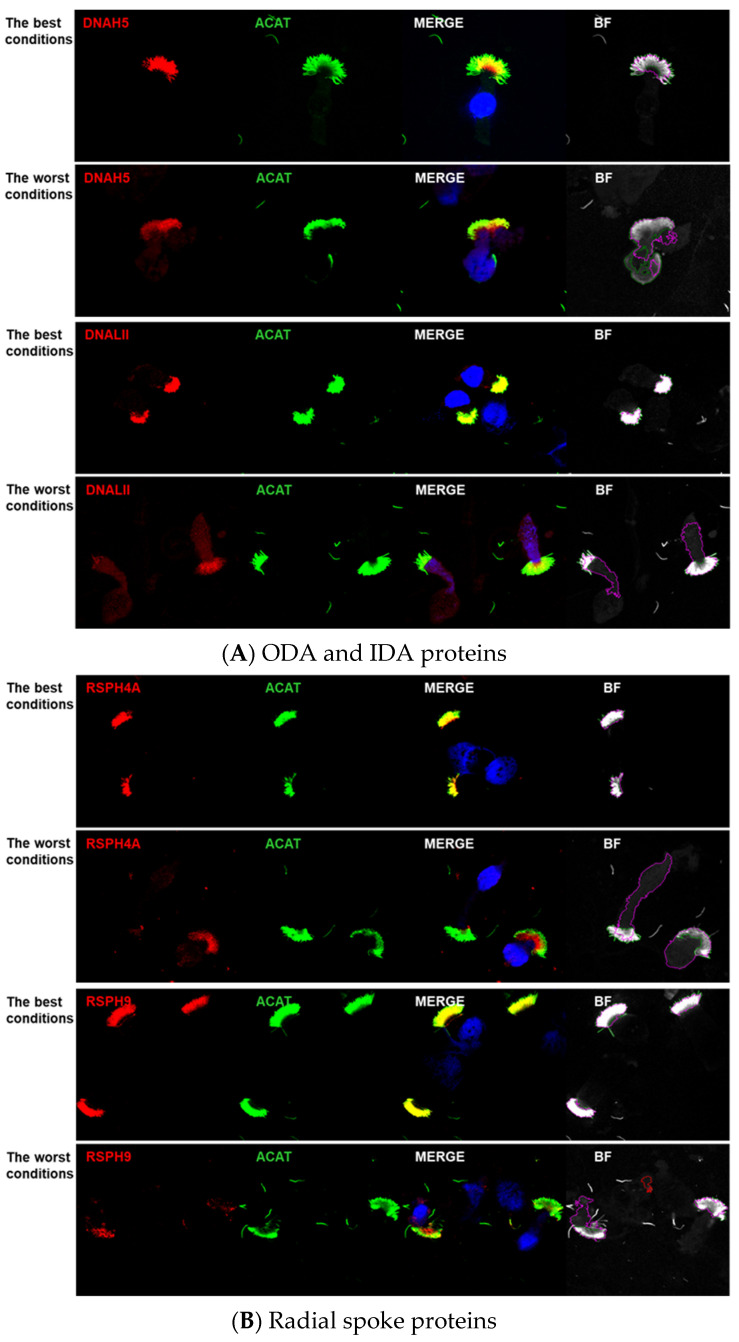
Representative IF images of various target proteins distribution (red) and the axoneme marker (green), at border storage conditions (−80 °C_28 d: the best conditions versus RT_28 d: the worst conditions). (**A**) ODA and IDA proteins; (**B**) RS proteins; (**C**) CP complex protein; (**D**) N-DRC protein; (**E**) MR proteins. Nuclei were stained with DAPI (blue).

**Figure 2 cells-14-01236-f002:**
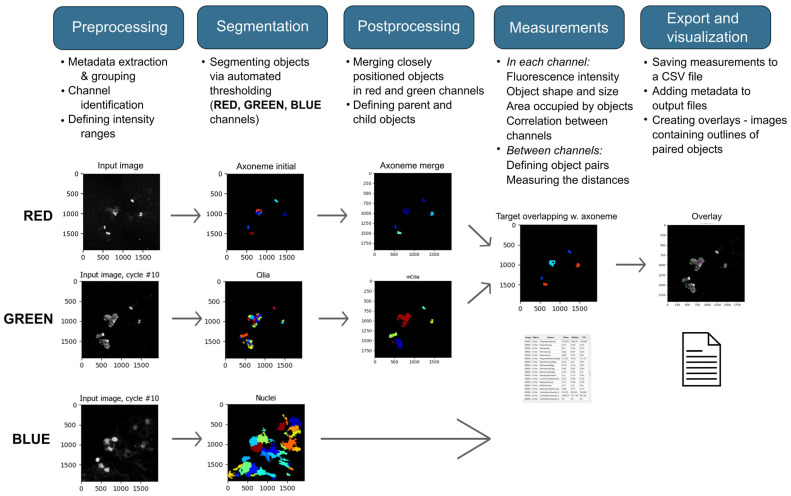
Custom CellProfiler pipeline.

**Figure 3 cells-14-01236-f003:**
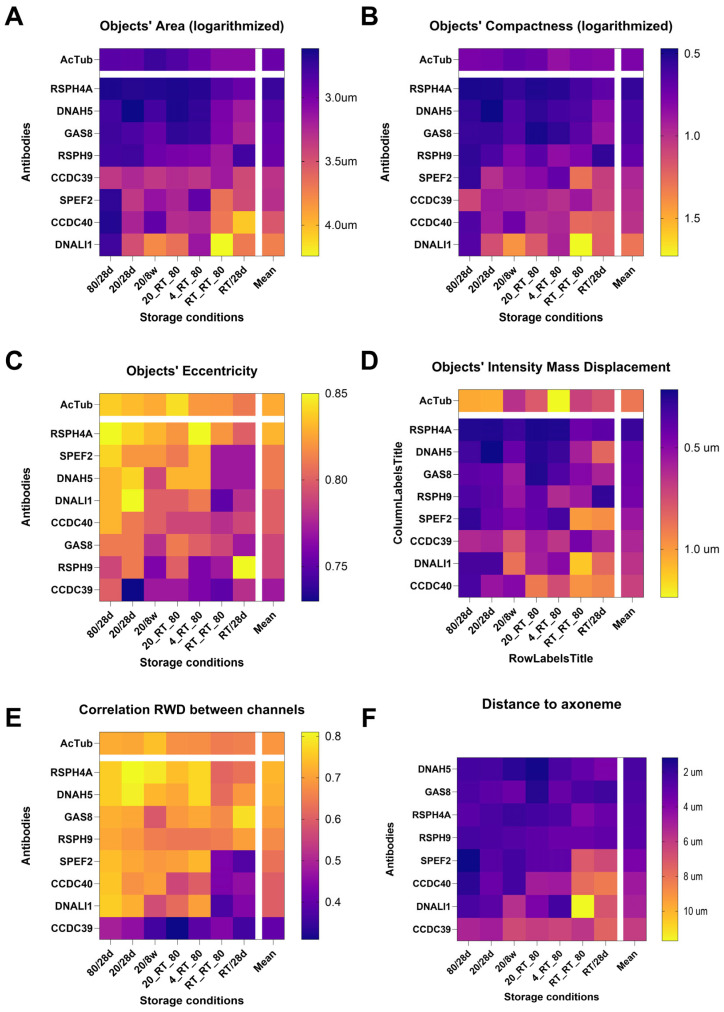
Comparison of the red objects’ characteristics. (**A**) Area; (**B**) compactness; (**C**) eccentricity; (**D**) intensity mass displacement; (**E**) RED-GREEN correlation RWD; (**F**) distance to axoneme marker. The parameters (values represented by colors in the heatmap) were calculated separately for each tested antibody and storage condition. Heatmap legends (see column to the right of each heatmap) are ordered according to the parameter’s values, indicating highest compatibility with the axoneme marker; the order of antibodies is according to their averaged parameter value across all conditions. The order of storage conditions (on X axis) is identical in A-F. For graphs in A and B, values have been logarithmized to improve the heatmaps’ utility and facilitate interpretation. Calculations were made for green objects pairing with only 1 red object.

**Table 1 cells-14-01236-t001:** Storage conditions.

	Group	Conditions (Temperature and Duration Time)		
1	−80_28 d (best conditions)	−80 °C 28 days		
2	−20_28 d	−20 °C 28 days		
3	−20_8 w	−20 °C 8 weeks		
4	−20/RT/−80	−20 °C 11 days	RT 3 days	−80 °C 14 days
5	4/RT/−80	4 °C 11 days	RT 3 days	−80 °C 14 days
6	RT/RT/−80	RT 11 days	RT 3 days	−80 °C 14 days
7	RT_28 d (worst conditions)	RT 28 days		

**Table 2 cells-14-01236-t002:** General performance ranks of slide storage conditions. The numbers in the table correspond to the order of conditions in the heatmap (1 being best, 7 being worst). For each of the analyzed parameters, slide storage conditions were ranked according to their overall performance (as indicated by the conditions’ order in each heatmap). This overall performance rank was calculated as the mean of all parameters; colors of the bars below the table indicate the performance level (green—very good, yellow—average, red—poor).

*Storage Condition*	−80_28 d	−20_28 d	−20/RT/-80	−20_8w	4/RT/−80	RT/RT/−80	RT_28 d
Mean_mTest_Correlation_RWC_RED_GREEN	1	2	5	4	3	6	7
Mean_mTest_AreaShape_Compactness	1	2	3	4	5	6	7
Mean_mTest_AreaShape_Area	1	2	4	3	5	6	7
Mean_mTest_Area_Eccentricity	1	2	3	5	4	6	7
Mean_mTest_Intensity_Mass_Displacement	1	3	2	5	4	7	6
Mean_mTest_Distance_Centroid_mCilia	1	2	4	3	5	7	6
*Overall Performance Rank:*	1.0	2.2	3.5	4.0	4.3	6.3	6.7


**Table 3 cells-14-01236-t003:** Overall performance of analyzed antibodies. For each of the analyzed parameters, antibodies were ranked according to the mean parameter’s value for all storage conditions (see rightmost columns in the heatmaps). The overall performance rank was calculated as the mean of all parameters. Colors of the bars below the table indicate the performance (green—very good, yellow—average, red—poor). Calculations were made for green objects pairing with only 1 red object.

	RSPH4A	DNAH5	GAS8	RSPH9	SPEF2	CCDC40	DNALI1	CCDC39
Mean_mTest_Correlation_RWC_RED_GREEN	1	2	3	4	5	6	7	8
Mean_mTest_AreaShape_Compactness	1	2	3	4	5	6	7	8
Mean_mTest_AreaShape_Aarea	1	2	3	4	5	6	7	8
Mean_mTest_Area_Eccentricity	3	6	4	2	1	7	5	8
Mean_mTest_Intensity_Mass_Displacement	1	2	3	4	5	8	7	6
Mean_mTest_Distance_Centroid_mCilia	4	1	2	3	5	6	7	8
*Overall Performance Rank:*	1.8	2.5	3.0	3.5	4.3	6.5	6.7	7.7


**Table 4 cells-14-01236-t004:** Comparison of the antibody performance ranks obtained for all storage conditions versus storage conditions excluding prolonged RT storage. Colors of the fields indicate the antibody performance (green—very good, yellow—average, red—poor). Calculations were made for green objects pairing with only 1 red object.

Antibody	All Storage Conditions	All Conditions Except Prolonged RT
RSPH4A	1.8	1.5
DNAH5	2.5	2.2
GAS8	3.0	4.2
RSPH9	3.5	4.5
SPEF2	4.3	4.2
CCDC40	6.5	5.7
DNALI1	6.7	6.0
CCDC39	7.7	7.8

## Data Availability

The original raw data presented in the study are available in Zenodo under https://doi.org/10.5281/zenodo.15463498. A preprint of the manuscript has been submitted to MedrXiv (MEDRXIV/2025/330648).

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
