# Peer review of "Systematic Comparison of Temperature Effects on Antibody Performance via Automated Image Analysis: A Key for Primary Ciliary Dyskinesia Diagnostic"

_cells, 2025, doi:10.3390/cells14161236_

Round 1

Reviewer 1 Report

Comments and Suggestions for Authors

This study discusses the diagnosis of Primary Ciliary Dyskinesia (PCD) by investigating the subcellular localization of proteins involved in the disease. The authors examined multiple antibodies under various storage conditions and analyzed how these conditions affect immunofluorescence staining. This research could be valuable for diagnostic purposes, as storage conditions may significantly impact sample quality for PCD diagnosis. Additionally, the use of whole-slide scanning and automated image analysis enhances the reproducibility of the results. I have a few suggestions, listed below:

  1. Briefly describe the functions of the following proteins: DNAH5 – ODA, DNALI1 – IDA, RSPH4A and RSPH9 – RS, GAS8 – N-DRC, CCDC39 and CCDC40 – MR, SPEF2 – CP complex.
  2. Provide brief explanations of technical terms such as "Mass Displacement" and "RED_GREEN correlation" to aid readers who may not be familiar with Cell Profiler.
  3. Increase the font size in Supplementary Figures S1 and S2, as it is currently too small to read easily.

Author Response

Dear Reviewer,

Thank you very much for taking the time to review this manuscript. Please find the detailed responses below and the corresponding corrections in track changes in the re-submitted files. 

Reviewer 1 comments: 

This study discusses the diagnosis of Primary Ciliary Dyskinesia (PCD) by investigating the subcellular localization of proteins involved in the disease. The authors examined multiple antibodies under various storage conditions and analyzed how these conditions affect immunofluorescence staining. This research could be valuable for diagnostic purposes, as storage conditions may significantly impact sample quality for PCD diagnosis. Additionally, the use of whole-slide scanning and automated image analysis enhances the reproducibility of the results.

Thank you very much for the positive words about our work!

Comment 1: Briefly describe the functions of the following proteins: DNAH5 – ODA, DNALI1 – IDA, RSPH4A and RSPH9 – RS, GAS8 – N-DRC, CCDC39 and CCDC40 – MR, SPEF2 – CP complex.

In the Introduction, we have included a section that briefly describes the function of each ciliary protein mentioned in your comment.

Comment 2: Provide brief explanations of technical terms such as "Mass Displacement" and "RED_GREEN correlation" to aid readers who may not be familiar with Cell Profiler.

The parameters were already described in the manuscript (in the Channel Parameters section of the Results). However, to make them stand out more and be more clear for readers not familiar with Cell Profiler, we have rewritten the paragraphs with the descriptions in order to make them more clear. The full descriptions are cited below:    

Intensity Mass Displacement parameter characterizes the distribution of the signal within the object. It measures the distance between the intensity-weighted center of the object and its geometric center: the higher the distance, the more asymmetric is the signal intensity distribution within the object.

RED_GREEN Correlation characterizes the correlation between the pixel intensities of the red channel and the green channel. It is an image-based (as opposed to object-based) parameter, which assesses how changes in the red signal intensity are linked to changes in the green signal at the same spatial locations within the image. The values of the RED-GREEN Correlation can range from -1 to 1. High positive values indicate that when the red is high (or low), the signal in the green channel tends to be high (or low) as well; values lower than 0 suggest a reverse correlation.

Comment 3: Increase the font size in Supplementary Figures S1 and S2, as it is currently too small to read easily.

We have increased the font size and figure size in Supplementary Figures S1 and S2 to improve their readability.

Quality of English Language:

(x) The English could be improved to more clearly express the research.
Response: We have edited the manuscript, particularly the Introduction and Results sections, to ensure that our study is thoroughly and clearly described; we have also corrected some punctuation issues.

We hope that these revisions fully address your comments. We are grateful for your time and expertise!

Sincerely,

Zuzanna Bukowy-Bieryllo

Reviewer 2 Report

Comments and Suggestions for Authors

This is a timely and well-designed study examining how slide storage conditions affect immunofluorescence (IF) staining in Primary Ciliary Dyskinesia (PCD) diagnostics. The authors systematically tested temperature and duration variables relevant to clinical workflows and used automated image analysis to strengthen their findings.

A key takeaway is that not all antibodies perform equally under suboptimal conditions—some, like those for RSPH4A, DNAH5, and GAS8, remain reliable, while others (CCDC39, CCDC40) are more vulnerable to room temperature exposure. These practical insights are especially useful for labs with limited freezing capabilities.

With minor clarifications in figures and methods, this work offers valuable guidance for improving the reliability of PCD diagnostics and merits publication.

Edits in the wording and puctuation may be needed but are minor.

Author Response

Dear Reviewer,

Thank you for your thorough review and for the positive comments on our manuscript. We have carefully considered your feedback and have revised the manuscript accordingly.  Please find the detailed responses below and the corresponding corrections in track changes in the re-submitted files. 

Comment 1: This is a timely and well-designed study examining how slide storage conditions affect immunofluorescence (IF) staining in Primary Ciliary Dyskinesia (PCD) diagnostics. The authors systematically tested temperature and duration variables relevant to clinical workflows and used automated image analysis to strengthen their findings.

A key takeaway is that not all antibodies perform equally under suboptimal conditions—some, like those for RSPH4A, DNAH5, and GAS8, remain reliable, while others (CCDC39, CCDC40) are more vulnerable to room temperature exposure. These practical insights are especially useful for labs with limited freezing capabilities.

We greatly appreciate your kind words about the design of our study and your acknowledgment of the practical applications of our findings, especially for labs with limited resources.

Comment 2: With minor clarifications in figures and methods, this work offers valuable guidance for improving the reliability of PCD diagnostics and merits publication.

Edits in the wording and punctuation may be needed but are minor.

Following your suggestions, we have made the following changes to the manuscript:

  • Figure 3 and Legend: We have replaced Figure 3 with a newer version and revised the legend to make it clearer and more informative.
  • Introduction and Results: We have edited these sections to ensure that our study is thoroughly and clearly described.
  • Wording and Punctuation: We have also carefully reviewed the entire manuscript to correct minor punctuation and wording issues.
  • Materials and Methods: In line with the requirements of the Editorial Office, we have moved the Materials and Methods section to appear after the Introduction. This section now includes a clear statement directing readers to the supplementary file for the complete Supplementary Materials and Methods.

We believe these revisions have significantly improved the manuscript. We are grateful for your time and expertise.

Sincerely,

Zuzanna Bukowy-Bieryllo